Estimating the impact of new high seas activities on the environment: the effects of ocean-surface macroplastic removal on sea surface ecosystems

Spencer Matthew 1
Culhane Fiona 1 2
Chong Fiona 3 4
Powell Megan O. 5
Roland Holst Rozemarijn J. 6
Helm Rebecca rh1203@georgetown.edu 7
1 School of Environmental Sciences, University of Liverpool , Liverpool , United Kingdom
2 School of Biological and Marine Sciences, University of Plymouth , Plymouth , United Kingdom
3 Energy and Environment Institute, University of Hull , Hull , United Kingdom
4 Biological and Marine Sciences, University of Hull , Hull , United Kingdom
5 University of North Carolina at Asheville , Asheville , NC , United States of America
6 Durham Law School, Durham University , Durham , United Kingdom
7 Earth Commons Institute, Georgetown University , Washington , DC , United States of America
Siuda Amy
Electronic publication date: 2023 Apr 27
Publication date: 2023
Volume: 11
Electronic Location ID: e15021
Received 2022 Nov 29; Accepted 2023 Feb 17
Copyright: ©2023 Spencer et al.
Copyright year: 2023
Copyright holder: Spencer et al.
License: This is an open access article distributed under the terms of the Creative Commons Attribution License, which permits unrestricted use, distribution, reproduction and adaptation in any medium and for any purpose provided that it is properly attributed. For attribution, the original author(s), title, publication source (PeerJ) and either DOI or URL of the article must be cited.
License URL: https://creativecommons.org/licenses/by/4.0/

Keywords: Neuston, Plastic, BBNJ, Ocean cleanup, United nations, High seas, Areas beyond national jurisdiction, The ocean cleanup, Netherlands, Environmental impact assessments

Funding: The University of Liverpool Institute for Risk and Uncertainty “The Ocean Cleanup Symposium 2019” The United States National Aeronautics and Space Administration 80NSSC21K0857 This work was supported by the University of Liverpool Institute for Risk and Uncertainty “The Ocean Cleanup Symposium 2019,” and the United States National Aeronautics and Space Administration (No. 80NSSC21K0857). The funders had no role in study design, data collection and analysis, decision to publish, or preparation of the manuscript.

==============================
The open ocean beyond national jurisdiction covers nearly half of Earth’s surface and is largely unexplored. It is also an emerging frontier for new types of human activity. Understanding how new activities interact with high seas ecosystems is critical for our management of this other half of Earth. Using The Ocean Cleanup (TOC) as a model, we demonstrate why it is important to account for uncertainty when assessing and evaluating impacts of novel high seas activities on marine ecosystems. TOC’s aim is to remove plastic from the ocean surface by collecting it with large nets. However, this approach also results in the collection of surface marine life (neuston) as by-catch. Using an interdisciplinary approach, we explore the social-ecological implications of this activity. We use population models to quantify potential impacts on the surface ecosystem; we determine the links between these ecosystems and society through an ecosystem services approach; and we review the governance setting relevant to the management of activities on the high seas. We show that the impact of ocean surface plastic removal largely depends on neuston life histories, and ranges from potentially mild to severe. We identify broader social-ecological implications that could be felt by stakeholders both beyond and within national jurisdiction. The legal framework applicable to TOC’s activities is insufficiently specific to address both the ecological and social uncertainty we describe, demonstrating the urgent need for detailed rules and procedures on environmental impact assessment and strategic environmental assessment to be adopted under the new International Agreement on the conservation and sustainable use of marine biological diversity of areas beyond national jurisdiction which is currently being negotiated.

Introduction

The high seas lie beyond national jurisdiction, covering nearly 50% of the Earth’s surface and constituting over 64% of the ocean by area. The ecological diversity of the high seas, and our reliance on it, is complex and poorly defined. This is especially true for the high sea ocean surface, which connects diverse ecosystems (Helm, 2021) and regulates ocean atmosphere exchange (McGillis et al., 2004). The ocean surface is also the front line for anthropogenic impacts from climate change, ship traffic, oil spills, and plastic pollution. These impacts occur in the same thin water layer as surface-associated marine life, termed neuston. We know very little about neuston or the impact human activity may have on the neuston ecosystem, although neuston are thought to be important in biogeochemical cycling and marine food webs, and to be threatened by pollution and climate change (Zaitsev, 1997). Due to its relative inaccessibility, the ocean’s surface is an exceptional study system for the legal, social, and environmental challenges facing policy makers attempting to ensure a sustainable future for the high seas.

One human impact on the open ocean that has particularly captured public imagination is plastic pollution (Kaiser, 2010), and no place is more infamous than the Great Pacific Garbage Patch (GPGP) (Kostigen, 2008). Plastic pollution negatively affects many coastal species (Gall & Thompson, 2015), but in the open ocean, the impact of plastic on marine life is complex and poorly studied, especially for the GPGP (Boerger et al., 2010; Wedemeyer-Strombel et al., 2015; Goldstein & Goodwin, 2013; Churchill, Valdés & Ó Foighil, 2014). Plastic may be ingested (Boerger et al., 2010; Wedemeyer-Strombel et al., 2015; Goldstein & Goodwin, 2013), and serve as a vector for invasive species (Goldstein, Rosenberg & Cheng, 2012), but it may also provide breeding habitat (Goldstein, Rosenberg & Cheng, 2012), and substrate for rafting organisms. Neustonic species that do not directly rely on plastic but that have low atmospheric drag, may, like ocean-surface plastic, be concentrated in the GPGP and coexist there (Egger et al., 2021).

As a likely result of public attention, several organizations are now dedicated to cleaning up ocean-surface plastic, the most prominent of which is The Ocean Cleanup (TOC). Plastic cleanup is generally considered as beneficial to the environment due to the dangers that plastics pose to marine life (Gall & Thompson, 2015). However, so little is known about the specifics of high sea ecosystems that this premise is worth closer scrutiny. There is a risk that TOC and similar initiatives could become part of an “innovation hype cycle”, meaning that their technology may not offer the best plastic catch rate for the effort, and could have unintended environmental consequences (Falk-Andersson, Haarr & Havas, 2020). TOC’s general proposal is to deploy a fleet of paired ships, each pair dragging a large U-shaped net between them to collect plastic, which will then be harvested and transported to shore. This kind of cleanup device is inspired by purse seine nets and technology used to trap floating oil, algae, and jellyfish (Brambini et al., 2017), and serves to concentrate floating objects until they can be harvested. As a result, there is a risk that neustonic animals and other marine life are also trapped in these nets, and this may have implications for the high sea ecosystem. TOC has commissioned two independent Environmental Impact Assessments (EIA) of their cleanup system. TOC’s first EIA omitted the neustonic ecosystem from the assessment (CSA Ocean Sciences, Inc., 2018), and the second EIA flagged potential impact on neuston as an area of concern (CSA Ocean Sciences, Inc., 2021).

This new activity on the high seas and the resultant questions around the interaction of surface-plastic cleanup technology and neuston exemplify the ecological, scientific, social, and political challenges facing areas beyond national jurisdiction. Understanding and estimating the impact of human activities on the high seas, as well as the potential consequences thereof, are a prerequisite for effective conservation and management. Yet, as we show, the relative ignorance of open-ocean biodiversity and ecology requires a fundamentally different approach to estimating high seas impacts than that applied to habitats closer to shore.

In this article, we examine the challenges posed by surface-plastic cleanup on the high seas from three perspectives: first, we model the impact TOC’s technology could have on neuston; second, we examine the societal benefits of neuston in terms of ecosystem services; and third, we identify the political and legal implications of the deployment of plastic-catching technologies in areas beyond national jurisdiction. We show that the effects of cleanup on neuston populations could plausibly be anywhere between negligible and extremely substantial, that neuston provide valuable ecosystem services, and that the international legal framework applicable to TOC’s activities is ambiguous and dependent on data that are not currently available to inform the content of legal obligations. We argue that our lack of knowledge about high seas ecology severely limits our ability to adequately assess human impacts on ecosystems and ecosystem services, and that the current legal framework does not provide robust tools to deal with this uncertainty or to weigh the different potential risks involved. This underlines the importance of adopting detailed rules and procedures for environmental impact assessment and strategic environmental assessment under the new International Agreement on the Conservation and Sustainable use of Marine Biological Diversity of Areas Beyond National Jurisdiction (BBNJ).

Methods

Model

Assumptions and modelling approach

We consider a deterministic model for the effects of floating macroplastic and ocean cleanup on a single species of neuston in continuous time, ignoring spatial and life history structure and seasonal or other variation in parameter values. Our aim is to provide a qualitative understanding of the system, focusing on equilibrium behaviour in order to inform long-term management strategies for plastic in the oceans. Little is known about interspecific interactions in the neuston, so a multispecies model is currently beyond our capabilities. There is recent evidence of interspecific competition in the neuston from stable isotope studies (Albuquerque et al., 2021). However, the general claim that interspecific interactions are weaker than intraspecific interactions (Mutshinda, O’Hara & Woiwod, 2009) appears to be supported by specific models for aquatic systems (e.g., Lindegren et al., 2009; Forsblom et al., 2021) to the extent that it is built into priors for multispecies models (Ward, Marshall & Scheuerell, 2022). We therefore model only a single species. Additionally, we include only floating macroplastics (particles with size > 0.5 cm; from here on simply plastics), rather than other fractions such as microplastics, because macroplastics are the target of current cleanup efforts.

Our model satisfies the postulate of parenthood, that every living organism has arisen from at least one parent of like kind (Hutchinson, 1978, p. 1), and thus ignores immigration. The neuston is in fact an open system. However, ignoring immigration allows us to frame the problem in terms of the niche structure of a neuston species. The fundamental niche of a species is defined as the set of environmental conditions under which the species can persist indefinitely, and “indefinite persistence” is generally taken to be in the absence of immigration (Holt, 2009). Within the fundamental niche, the proportional population growth rate, ignoring immigration, represents the population-level response of a species to its environment (Maguire, 1973). Such a definition also makes sense for ecosystem functions or services that depend on production, but not those that depend on abundance or biomass. In addition, any cleanup programme aiming to achieve a large reduction in total floating macroplastic would have to operate over a large area, for which it is likely that external inputs would be small compared to the effects of internal dynamics. We focus here on true neuston, which remain at the surface throughout the diurnal cycle. There are also important groups of organisms facultatively associated with the ocean surface, but undergoing diel migration (Hempel & Weikert, 1972). The equilibrium behaviour of a model ignoring diel migration may be a reasonable approximation for the long-term effects of cleanup on such organisms.

We assume that intraspecific interactions can be described by logistic density dependence. The logistic model is widely used, and is the simplest density-dependent model satisfying the postulate of parenthood (Hutchinson, 1978, p. 4). Furthermore, logistic density dependence has the convenient property that we can study effects on equilibrium neuston density relative to its value in the absence of cleanup, without data on the strength of intraspecific density dependence. This is important, given the scarcity of demographic data on neuston populations. We initially describe a model in which plastics can affect the proportional population growth rate of neuston. However, there are very few data on the population-level effects of plastics on ocean organisms. We therefore assume in subsequent analysis of the effects of cleanup (which act through removal of both neuston and plastics) that the effect of plastics on neuston is zero. Assuming no effect of plastics on neuston is conservative with respect to the possible net negative effect of cleanup. Furthermore, the most relevant tradeoff is between negative effects of cleanup on neuston and positive effects on other ocean organisms, rather than between negative and positive effects on neuston.

We model the dynamics of plastic concentration at the ocean surface with a single compartment representing buoyant macroplastics with a constant input rate and a constant natural loss rate per unit plastic concentration. Although models with multiple compartments such as those found in Koelmans et al. (2017) and Lebreton, Egger & Slat (2019) are needed to study the global dynamics of ocean plastic, the buoyant macroplastics compartment is the one most relevant to the effects of ocean cleanup on neuston.

Initial model description

Here, we describe our initial model, including an effect of plastics on the proportional population growth rate of neuston. Let n be neuston density (dimensions ML−2; throughout we use the standard symbols M, L and T to refer to the dimensions mass, length and time respectively), let p be plastic density (dimensions ML−2) and let t be time (dimensions T). We use a logistic population growth model for neuston, coupled with an input–output model for plastic dynamics:

(1) dndt=a1n+a2n2+a3np−c1kn

(2) dpdt=b1−b2p−c2kp.

The structure of the model is summarized in Fig. 1. In the neuston dynamics Eq. (1), a1 denotes neuston proportional population growth rate at low density (dimensions T−1) and a2 denotes the effect of neuston density on neuston proportional population growth rate (dimensions M−1L2T−1). We write the logistic neuston population growth equation as a second-order Taylor polynomial approximation around zero (Lotka, 1956, p. 65) with a1 > 0 and a2 < 0. In the absence of plastic and cleanup the population will increase when rare, and will have carrying capacity −a1/a2. The parameter a3 denotes the effect of plastic on neuston proportional population growth rate (dimensions M−1L2T−1). The sign of this parameter is unknown: it is possible that plastic has a positive effect on neuston proportional population growth rate (for example, some forms of plastic may provide substrate for attachment of eggs of some neuston species) (Goldstein, Rosenberg & Cheng, 2012). The positive parameter k denotes the effort devoted to ocean cleanup, measured in some convenient way such as energy, money or area swept per unit time (denoted [effort]T−1), and the positive parameter c1 denotes the rate of neuston removal per unit effort of cleanup (dimensions [effort]−1). We do not include an external input of neuston, as explained above.

Figure 1 Structure of the model defined by Eqs. (1) and (2).

The effect of plastic on neuston population growth (dashed arrow) is assumed to be zero from the ‘Relationship between equilibrium scaled plastic and neuston densities under cleanup when plastic has no direct effect on neuston’ section onwards.

In the plastic dynamics Eq. (2), the positive parameter b1 denotes external input of macroplastics into the open ocean (dimensions ML−2T−1), through routes such as transport from rivers via coastal waters (Lebreton, Egger & Slat, 2019). The positive parameter b2 denotes the natural loss rate of macroplastics from the layer of the ocean affected by cleanup (dimensions T−1). This is thought to occur mainly through fragmentation into microplastics (Koelmans et al., 2017; Lebreton, Egger & Slat, 2019). The positive parameter c2 denotes the rate of macroplastic removal per unit effort of cleanup (dimensions [effort]−1).

Full details of model analysis are given in the Supplemental Information Section S1.

Relationship between equilibrium scaled plastic and neuston densities under cleanup when plastic has no direct effect on neuston

We now make the simplifying assumption (justified in the ‘Assumptions and modelling approach’ section) that plastic has no effect on neuston proportional population growth rate (i.e., a3 = 0) and study the relationship between scaled plastic and neuston densities at equilibrium, relative to their values in the absence of cleanup. We treat scaled plastic density as under our control through some management strategy that determines cleanup effort, and examine how this will affect neuston. Let n∗ denote neuston concentration as a fraction of its equilibrium value in the absence of cleanup, and p∗ denote plastic concentration as a fraction of its equilibrium value in the absence of cleanup.

Under the assumption of no plastic effect on neuston, we can write the scaled equilibrium neuston density as a function of scaled equilibrium plastic density: (3) n∗p∗= max0,1−1p∗−1Π,

where the dimensionless parameter Π=b2a1c1c2 is the ratio of natural loss rate of macroplastics to neuston proportional population growth rate at low density, times the ratio of cleanup efficiencies. Thus a neuston population will be most affected if it has slow growth relative to the natural plastic loss rate (b2/a1 large), and if the cleanup strategy removes neuston at a high rate relative to plastic (c1/c2 large).

Parameter values

Here, we summarize the plausible ranges of the parameters b2, a1 and c1/c2 that we considered. Full details are given in the Supplemental Information. Estimates of the natural loss rate of plastic b2 vary widely, with differences in model assumptions making an important contribution to this variation. We considered the range 0.03a−1 to 1.26 a−1 (throughout, we use a−1 to denote units of per year). There is little information on proportional population growth rates at low density (a1) for neuston. We therefore used an allometric approach based on body size, which suggested the range 1.08a−1 to 63.52 a−1 for small neuston species, and the range 0.08a−1 to 4.75 a−1 for large neuston species. Little is known about the efficiency of neuston removal relative to plastic removal (c1/c2). Since neuston and floating plastic overlap in size and occur in the same location, 1 is a plausible value for this ratio. However, other values are not implausible, and we therefore considered the range [1/10, 10].

Visualization of model behaviour

Equation (3) shows that the relationship between scaled equilibrium neuston density and scaled equilibrium plastic density is determined entirely by the dimensionless parameter Π=b2a1c1c2. We therefore calculated the range of possible values of Π for small and large neuston species from the ranges for b2, a1 and c1/c2 (‘Parameter values’ section). We plotted the envelope of possible relationships between the proportion of neuston remaining (n∗) and the proportional reduction in plastic (1 − p∗) for small and large neuston species. To understand how this relationship depends on the underlying parameters b2, a1 and c1/c2, we plotted the relationship between n∗ and 1 − p∗ for five logarithmically-spaced values of one parameter at a time, spanning the plausible range of values, and holding the other two parameters at their geometric midpoints. We show in Supporting Information S3, that effects of cleanup on neuston density are likely to occur on a time scale of months to decades after the start of a cleanup programme.

Ecosystem services

Ecosystem services were identified following the approach used in Culhane et al. (2018), which identified all the links between the marine ecosystem and ecosystem services it supplies, using defined ecosystem component and ecosystem service typologies. The ecosystem components defined in that study are made up of a habitat and an associated biotic group. From that typology, the neuston populations considered here fit into the ‘zooplankton’ and ‘macroalgae’ biotic groups in the surface of the ‘oceanic waters’ habitat. In this work, we refer to them as zooneuston and phytoneuston. Links from the neuston were then made to ecosystem services they supply using the typology of marine ecosystem services (Culhane et al., 2019; Culhane et al., 2018), which was originally adapted for marine ecosystems from the Common International Classification of Ecosystem Services (CICES) v4.1 typology (Haines-Young & Potschin, 2013). This typology defines three broad categories of service, including provisioning, regulation and maintenance and cultural, with a total of 33 individual marine ecosystem service types. This typology includes both services that have a marketable value (e.g., seafood or raw materials) and services that are more intangible but nevertheless contribute to human wellbeing (e.g., aesthetic or existence values).

Due to the breadth of service types, specific links between neuston and an ecosystem service were identified where one or more of three criteria were met, depending on what was appropriate given the nature of the service type. Firstly, a link was identified where there was evidence of direct use e.g., for the Raw materials service, a link would be identified if there is evidence that a part of the neuston is harvested and used as a raw material. Secondly, a link was identified where functions of the neuston would lead to the supply of a service, based on ecological knowledge. An example of this is for the Waste treatment service. Neuston functional feeding groups include suspension, boring, detritus and scavenging modes (Thiel & Gutow, 2005), meaning they have good capacity to breakdown, remove and bioremediate organic and other waste from the ocean surface. Thirdly, a link was identified where there is evidence for potential use where this is appropriate for the service, for example, under the Genetic materials service, bioprospecting for medicinal or industrial properties that have not yet been discovered or extracted. Evidence came from ecological literature on the neuston (e.g., to find relevant functions), other literature (e.g., biochemical journals that document compounds used in medicine that are derived from neuston), and other internet sources (e.g., those that demonstrate use of neuston for artistic inspiration) See the Supplemental Information for more details.

Two types of link were identified as described in Culhane et al. (2018). Direct links are given where a service is supplied directly within the habitat e.g., waste bioremediation that occurs on the ocean surface (though the benefits of this may extend beyond this habitat). Indirect links are supplied in another habitat by the same population of organisms that is supported by oceanic waters. For example, Velella velella that live in oceanic waters can be washed into coastal areas, transferring a large amount of organic material to coastal and terrestrial environments supplying the Sediment nutrient cycling service in these habitats (Betti et al., 2017; Purcell, Clarkin & Doyle, 2012); eels found in the neuston of oceanic waters are the same individuals that are found in freshwaters and contribute to a number of ecosystem services, such as Seafood and Cultural heritage (Norfolk Coast Partnership, 2020). These services are being supplied directly in coastal or freshwater habitats, but oceanic habitats contribute to supporting their supply. This method recognises that, although we are considering neuston present in the open ocean, these same populations are directly connected to habitats beyond the open ocean, and are supplying services in other habitats. Indirect services were not indicated if the service was also supplied directly. Services identified were not quantified, and thus, as long as one of the three criteria above were fulfilled, the service was counted as being supplied by neuston in oceanic waters.

Legal

The legal perspective relied on legal doctrinal methodology to first identify the law applicable to TOC’s activities, as well as the gaps therein, on two different levels: the obligations of the Netherlands as the responsible state under international law; and how these obligations of the state are ‘translated’ into specific obligations on TOC under the 2018 Agreement concluded between the Dutch government and TOC.1 The focus is on the obligations relating to the protection of the marine environment. Secondly, the legal relevance of uncertainty as to both the risks and benefits involved in operating a new technology in a sensitive environment were discussed, revealing how legal rules and standards presuppose the availability of at least some (environmental) data and knowledge.

Results

Model

Possible outcomes of a cleanup programme range from negligible equilibrium effects on both small and large neuston even for large reductions in equilibrium plastic to very substantial equilibrium reduction in neuston even with small reduction in equilibrium plastic (Fig. 2: grey envelopes, with negligible effects in the top right corner and large reductions in the bottom left corner). For a given proportional reduction in plastic, the proportion of neuston remaining increases as neuston proportional population growth rate a1 increases (Figs. 2A and 2B; stronger colours represent larger a1), decreases as the natural loss rate of plastic b2 increases (Figs. 2C and 2D; stronger colours represent larger b2), and decreases as the efficiency ratio c1/c2 increases (Figs. 2E and 2F; stronger colours represent larger c1/c2). For given values of b2 and c1/c2, the equilibrium proportion of neuston remaining tends to be smaller for large than for small neuston, because the plausible range of a1 contains smaller values for large than for small neuston (Figs. 2B, 2D and 2F versus 2A, 2C and 2E). These results agree with intuition: we would expect neuston species with lower proportional population growth rates to be less able to absorb additional mortality from cleanup; if the natural loss rate of plastic is larger, more cleanup effort will be needed to achieve a given proportional reduction in plastic; and if the efficiency ratio is higher, a given cleanup effort will remove more neuston relative to plastic.

Figure 2 Relationship between equilibrium proportion of neuston remaining (n∗) and equilibrium proportional reduction in plastic (1 − p∗) for small (A, C, E) and large (B, D, F) neuston species, and for varying parameter values.

On each panel, the grey envelope encloses the set of possible relationships. In A and B, lines represent the relationship as neuston proportional population growth rate at low density a1 (units a−1) varies over its plausible range of values (which differs for small and large neuston), with b2 and c1/c2 held at their geometric midpoints. In c and d, lines represent the relationship as natural loss rate of plastic b2 (units a−1) varies over its plausible range of values, with a1 and c1/c2 held at their geometric midpoints. In E and F, lines represent the relationship as the efficiency ratio c1/c2 (dimensionless) varies over its plausible range of values, with a1 and b2 held at their geometric midpoints. On each panel, stronger colours represent increasing logarithmically-spaced values of the varying parameter, and the middle line corresponds to the geometric midpoint of the plausible range for the parameter.

Ecosystem services

Ecosystem services of the neuston in the GPGP are poorly known, so we evaluated the services of neuston more broadly, as a proxy to understand potential ecosystem services that can be applied to neuston in the GPGP. We found that neuston in oceanic waters supply at least 28 services (20 services that have direct links, and eight that have only indirect links, out of a total of 33 possible services (Fig. 3, Supplemental Information for full details). Many of the services supplied by the neuston, either directly or indirectly, show that neuston facilitate connectivity between remote and accessible coastal, freshwater and terrestrial habitats. For example, neuston are an important food source for marine predators such as turtles (Witherington, 2002; Revelles et al., 2007), migratory birds such as the sooty shearwater, species of storm-petrel, shearwater (Ribic, Ainley & Spear, 1997), Phalaropes (DiGiacomo et al., 2002) and for commercially important fish species such as tuna (Thiebot & McInnes, 2020; D’Ambra et al., 2015) and hence provide regulation and maintenance services (Maintaining nursery population and habitats). Neuston also make a notable contribution to cultural services, such as Aesthetic, for example the artist Aaron Ansarov, who takes inspiration from neuston washed ashore by photographing live specimens of Physalia sp. (Davis, 2013).

Figure 3 Ecosystem services (ecological and societal benefits of neuston) provided by the neuston considered in this study.

There are three types of service: Provisioning, Regulation and maintenance, and Cultural. Direct (solid line) and indirect (dashed line) links are shown, where direct links are supplied directly in ocean surface habitats while indirect links are supplied in other habitats but supported by open ocean surface communities. Full details of links can be found in Tables S1–S2.

Legal Implications

TOC provides an interesting example of how technological developments and new types of activities are taking a growing variety of actors to the high seas, where they may come to interact with little-known ecosystems like neuston. The example of TOC thereby highlights a number of relevant regulatory and governance challenges. Firstly, it should be noted that TOC is a private actor, operating in areas beyond national jurisdiction (high seas). Under international law, the legal framework set out in the UN Convention on the Law of the Sea (UNCLOS) determines which state can do what, and where in the world’s oceans. As TOC is a legal entity incorporated under Dutch law, the Dutch Government has a general obligation under UNCLOS and general international law to ensure that activities under its jurisdiction and control do not cause harm to other states or to the marine environment, including in areas beyond national jurisdiction. This general obligation is not an obligation of result in the sense that the Netherlands is bound to prevent any harm from occurring, but rather an obligation of ‘due diligence’: a standard of care. There are a few core elements to this general obligation when it comes to the protection of the marine environment: the obligation to conduct a prior environmental impact assessment (EIA) when it cannot be excluded that an activity may cause significant harm to the marine environment, including marine biodiversity (a threshold that has been interpreted leniently by international courts and tribunals); the obligation to continuously monitor such risks; and take any (precautionary) measures necessary to prevent, control or minimise the risk of serious harm. Which measures exactly are ‘necessary’ and the standard of care required can only be determined on a case-by-case basis. This is exactly why it is essential to acquire adequate data and knowledge of the various ‘risks’ involved, before any detailed regulatory and governance decisions can be taken, or indeed challenged.

Due to the unique and unprecedented nature of TOC’s activities, there are no dedicated international or domestic regulations applicable to the operation of ‘cleanup systems’ to give further content to the general obligations in this respect. In order to ensure that TOC’s activities are at least conducted in accordance with general international law, the Dutch government entered into an Agreement with TOC on 8 June 2018 (hereafter ‘the Agreement’). This Agreement is applicable only between TOC and the Netherlands, and serves to ‘translate’ the core responsibilities and liabilities of the Netherlands under international law into binding obligations on TOC (Roland Holst, 2019). In other words; it is the instrument through which the Netherlands as the responsible state ‘regulates’ TOC’s activities, in accordance with the Netherlands’ obligation of due diligence under international law.

As far as the protection of the marine environment from (accidental) damage caused by the clean-up system is concerned, the Agreement requires TOC to take precautionary measures, and to remove any parts of the system from the high seas when they are no longer used. Precautionary measures are also required specifically for the protection of species in the area of operation, including the establishment of a monitoring plan, which is curiously limited to the first year of deployment on the high seas. Other than these ‘best efforts’ obligations, the Agreement does not set out any concrete environmental standards or obligations, nor does it differentiate between the operation of a single system and the envisaged scale-up. Noteworthy in particular is the fact that the need for an EIA is not mentioned anywhere in the Agreement.

TOC published an EIA on its own initiative in July 2018 before towing the first system to the high seas (CSA Ocean Sciences, Inc., 2018), and a second one in July 2021 for a new iteration of the system (CSA Ocean Sciences, Inc., 2021). Presumably for this reason and the fact that the initial EIA did not establish a risk of significant harm to the marine environment, the Agreement does not mention the need for an EIA anywhere. Nevertheless, this appears to be a lacuna. Whereas the 2018 EIA omitted neuston from the assessment, the 2021 EIA confirms that neuston may be the ecosystem and group of species potentially impacted the most. While initial trials of a single cleanup system are relatively small-scale, and therefore arguably not likely to pose ‘significant’ risks to the marine environment including neuston, future iterations of the system and/or the proposed scale-up to a fleet of bigger systems may significantly change the potential impacts in the future. Reasonable grounds to expect that significant harm may nevertheless occur could arise at a later stage of the project, in which case the Netherlands is required under UNCLOS and general international law to make sure these risks are (re)assessed and continuously monitored. If the neuston could furthermore be considered an important ‘rare and fragile ecosystem’, or even the habitat of ‘depleted, threated or endangered species’, this would raise the standard of care and precaution required vis-à-vis the neuston in accordance with the Netherlands’ obligations not only under UNCLOS, but also e.g., the Convention on Biological Diversity, and the new BBNJ Agreement.

Discussion

With the current state of knowledge, effects of plastic removal on neuston populations could plausibly be anywhere from negligible to very substantial. Three key parameters determine these effects: the maximum proportional population growth rate of neuston at low density; the natural loss rate of macroplastic; and the efficiency ratio of neuston removal to macroplastic removal. We outline below how the uncertainty in these parameters could be reduced. However, only the efficiency ratio is under human control. We showed that neuston directly provide important ecosystem services, and indirectly support services supplied by coastal, terrestrial and freshwater ecosystems. A technological intervention to tackle the problem of ocean-surface macroplastic pollution therefore involves balancing one environmental concern (impacts of plastic debris on the marine environment) against another environmental concern (impacts of the cleanup technology itself on the ecosystem). We argue below that this involves a novel type of balancing exercise, for which existing governance principles do not provide any concrete guidance.

All three of the key parameters determining the effects of ocean surface macroplastic removal on neuston populations are highly uncertain. For the maximum proportional population growth rate of neuston at low density, accurate estimates will require experimental measurement of vital rates under open-ocean-like conditions, for every stage in what may be a complex life cycle. Such measurements are challenging even for species that are relatively easy to culture (e.g., Goldstein & Steiner, 2020). For the natural loss rate of macroplastic, estimates from a year-long laboratory mesocosm experiment (Gerritse et al., 2020) are generally at the low end of the range used in our analyses. If correct, this may reduce the likelihood of adverse effects on the neuston for a given target reduction in ocean surface microplastic, because a smaller cleanup effort is required for a given proportional reduction in plastic. However, the rate of plastic input to the oceans may increase in the future without improvements in waste management (Jambeck et al., 2015), or decrease with plausible increases in recycling and incineration rates (Geyer, Jambeck & Law, 2017), so that future modelling may need to consider effects of cleanup on neuston under non-equilibrium macroplastic dynamics (Hohn et al., 2020). The efficiency ratio of neuston removal to macroplastic removal could in principle be measured in situ in field trials. This is the only one of the three key parameters that is under human control. There may be some scope for engineering developments that reduce this ratio. For example, physical characteristics such as atmospheric drag may influence the distributions of neuston species (Egger et al., 2021), and it might be possible to design cleanup devices that are least efficient at removing organisms with characteristics matching the most vulnerable species. However, until more data exist, this remains speculative.

Although remote, open ocean habitats are connected much more widely to different geographical regions, habitats and stakeholders, as evidenced by the range of ecosystem services they supply. There are important flows, not only from terrestrial/near-shore to open ocean habitats, but also from the open ocean via the neuston. The importance of the connection between remote habitats like the open ocean with global ecosystem functions and with near-shore coastal, terrestrial and freshwater habitats and their services must be emphasised when considering potential costs and benefits of impacts on these systems. The stakeholders of such ecosystems are far-reaching (Thurber et al., 2014) but lacking consideration under formal obligations. For example, critically-endangered European eels migrate to the Sargasso Sea to spawn, and impacts on the neuston community of this region would also potentially impact eels. In the North Pacific neuston are key prey items for loggerhead turtles and albatross (Helm, 2021). The neustonic ecosystem is also home to diverse larval fish and invertebrates (Whitney et al., 2021).

Unlike traditional exploitation activities, technological ‘solutions’ to environmental problems like TOC involve balancing one environmental concern (impacts of plastic debris on the marine environment) against another environmental concern (impacts of cleanup on the neuston and biodiversity). The objective either way is to protect and conserve the marine environment, but notions of ‘harm’ or ‘risk’ involved can be weighed very differently depending on stakeholders’ perspectives. This balancing act becomes even more complicated when (novel) activities interact with understudied ecosystems, meaning that uncertainty remains as to both the benefits of the technology addressing the target risk, and the potential risks involved in deploying the technology itself. Existing legal principles do not provide any concrete guidance or benchmarks in this connection. For example, the precautionary approach is typically applicable when uncertainty remains, yet, in the present context it may work both ways as to either allow the activity to proceed until more is known, or to restrict it, depending on how the short and long-term impacts and benefits are understood and weighed. Tools and principles such as ‘best available technology’, ‘best available science’ or ‘best environmental practices’ that are commonly used to give content to, for example, the precautionary approach and general due diligence obligations, are also of little help when there is no relevant ‘science’ or ‘practice’ available to compare it to. A new type of activity like TOC illustrates that the application of general environmental rules and principles presupposes at least some knowledge of a particular activity or technology, its consequences, risks, and possible alternatives. This issue arises not just in relation to neuston: the high seas are vastly understudied and these challenges may arise in relation to a variety of ecosystems. This is further magnified by the complexity of human impacts thereon.

Likewise, the impacts of plastics have only been studied for a small number of surface species, and range from potentially negative (fish), to potentially neutral (barnacles), to potentially positive (by providing substrate for reproduction) (the insect Halobates). Thus, plastic cleanup may benefit some species to the detriment of others. Our models demonstrate there may be substantial negative impact of cleanup on neuston populations, but naturally, in the absence of a negative ecosystem impact, plastic removal could have a positive environmental outcome.

In conclusion, we have shown that the potential effects of ocean surface and macroplastic removal on neuston populations are uncertain but potentially negative, and that the steps needed to reduce this uncertainty are clear in principle. Our approach highlights the critical need for more life history data for open-ocean species, and if limited data on these parameters exist, models of impact, like the one used here, should explicitly incorporate uncertainty. All impact assessments should also examine ecological services and ecosystem connectivity. In this connection there is an important role cut out for the new BBNJ Agreement, which sets out more detailed rules and procedures (including on public participation) for environmental impact assessment and strategic environmental assessment. New high seas activities like TOC that come into contact with understudied ecosystems for the first time pose both challenges and opportunities: they highlight the need to obtain further data and knowledge, including to give content to general legal obligations and to inform the broader governance framework for biodiversity beyond national jurisdiction, while emphasising the need for serious precaution as the exact scope and implications of human impacts on complex ecosystems remain only partly understood.

Supplemental Information

Supplemental Information 1 Code for generating figures to recreate our models

Click here for additional data file.

Supplemental Information 2 Ecosystem services

Click here for additional data file.

Supplemental Information 3 Mathematical model analysis

Click here for additional data file.

We thank the organizers and participants from the 2019 Ocean Clean-up Symposium for their contributions.

Additional Information and Declarations

Competing Interests

Author Contributions

Data Availability

1 Agreement between the State of the Netherlands and The Ocean Cleanup concerning the deployment of systems designed to clean up plastic floating in the upper surface layer of the high seas (The Hague, 8 June 2018) Staatscourant 2018 nr. 31907, 6 July 2018, available at https://zoek.officielebekendmakingen.nl/stcrt-2018-31907.html.

The authors declare there are no competing interests.

Matthew Spencer conceived and designed the experiments, performed the experiments, analyzed the data, prepared figures and/or tables, authored or reviewed drafts of the article, and approved the final draft.

Fiona Culhane conceived and designed the experiments, prepared figures and/or tables, authored or reviewed drafts of the article, ecosystem services analysis, and approved the final draft.

Fiona Chong conceived and designed the experiments, prepared figures and/or tables, authored or reviewed drafts of the article, ecosystem services analysis, and approved the final draft.

Megan O. Powell conceived and designed the experiments, performed the experiments, analyzed the data, prepared figures and/or tables, authored or reviewed drafts of the article, and approved the final draft.

Rozemarijn J. Roland Holst conceived and designed the experiments, prepared figures and/or tables, authored or reviewed drafts of the article, policy analysis, and approved the final draft.

Rebecca Helm conceived and designed the experiments, performed the experiments, analyzed the data, prepared figures and/or tables, authored or reviewed drafts of the article, and approved the final draft.

The following information was supplied regarding data availability:

The raw data and code are available in the Supplemental Files.

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
