# Peer review of "Estimating the impact of new high seas activities on the environment: the effects of ocean-surface macroplastic removal on sea surface ecosystems"

_PeerJ, doi:10.7717/peerj.15021_

## Round 0.1 · original submission · Minor Revisions

The reviewers make many valid comments which need to be addressed.

In addition, I would like to point out that care should be taken to address the tense changes throughout the manuscript. Specifically, the start of the Methods is in past tense and switches to present tense for the model description (all parts of 1.3). Again, in the Results, the model results (2.1) are in present tense and the rest of Results are in past tense.

The Methods and Results are not parallel. Consider rearranging, such that both flow in the same order of Ecosystem Services, Policy, Model or the other way around.

Finally, correct the "we thanks" in the Acknowledgements.

Reviewer 1 ·

Basic reporting

This is an interesting paper which attempts to quantify the potential effects of plastic removal efforts on open-ocean communities, and to shine a spotlight on ecosystem services and legal frameworks. In general it merits publication in some form. I feel that it would be made more powerful and palatable to policy makers if some of the model derivation were moved to supplement, and those approaches were described in more general terms.
* * *
Sweeping the parameters seems to go between extremes (high,low). The values swap over two orders of magnitude, skipping any mid values (1->63; 0.03->1.26; 0.1->10). (Is it realistic that plastic removal could approach anywhere near 50% efficiency?) Given these extremes, it is hard to interpret the results when there seems to be no consistent pattern to the models across the range of parameters tested.

For example, in figure 2, the proportion of neuston remains high across the board, then there are some intermediate values, but within the intermediate values, there is no clear pattern for High or Low parameters a,b,c. Represented are LLH, HHH, LHL. In figure 3, the clusters are equally disparate: LLH,HHH,LHL and HLH,LLL,HHL.

Another way to look at this is that in all scenarios there are potentially strong negative consequences from by-catch related to plastic removal efforts, but from the model I did not come away with a better understanding of the trade-offs for mitigating these harmful effects.

I don't know if it is possible, but it would be informative to create a box diagram to show some of the interactions and insights gained from the model.

The text describes an initial model and subsequent modifications to that model, but in the model description, it is hard (for me at least) to know where the boundary between another model come, or whether what is described is part of the derivation of that original model ("We now make the simplifying assumption...")
* * *
The focus of the paper is largely on modeling the potential reduction in neuston, while also touching on ecosystem services and relevant legal frameworks. In my reading, it starts with the assumption that plastic removal is bad, but let's see just how bad is it? Whatever my personal perspective, I think if I were associated with TOC, I would feel like this is not a very objective study. Besides rephrasing some of the more inflammatory language, it would be relatively straightforward to include a more quantitative evaluation of potential positive effects and overall efficiency of plastic removal. Can these few-ship operations actually make a different in the amount of plastic in the ocean? In the same way that other parameters are estimated, the authors could estimate parameters of removal relative to overall plastic abundance.

In other words, the overall evaluation of the situation should assimilate all factors: positive effects of plastic removal; negative effects of neuston removal; natural removal of plastic from the ecosystem; ecosystem services of neuston; legal considerations.
* * *
Style choices:
* The introduction is a bit rough. Some suggested edits are marked in the PDF. "impact" used a lot as both noun and verb. Consider alternatives.
* "TOC", not "The TOC"

Figure 4 is not very effective. It might be a lot easier to trace if the lines were replaced with a grid of filled/cross-hatched/empty cells. (See sketch in review PDF)

Experimental design

See above

Validity of the findings

See above

Additional comments

See above

Annotated reviews are not available for download in order to protect the identity of reviewers who chose to remain anonymous.

Reviewer 2 ·

Basic reporting

The paper by Helm and colleagues on the impact of plastic removal devices on neuston community in the high seas and the regulations that should be provided is clear and is answering important questions. Plastic removal in the oceans is becoming a general trend supported by private compagnies for which no assessment has been provided. Eventhough these cleaning system might have raised from well-meaning people, impacts on neuston can have devastating consequences that need to be addressed, even more in the high seas where lwas are currently lacking.The context and data provided in the paper are rather clear and well presented. Nevertheless the presentation of the neuston community and their importance should be more explicit and highlighted. This paper is very important and should be published after some corrections.

Experimental design

The research question are well stated and the modelling approach robust. The authors should make sure that each untis and parameters in well define and follow accepted format. They express their density in term per cubic liter, but most data on plastic are express per square kilometers making numbers looking a lot more important than they are actually. SOmething about this should be added.

Validity of the findings

The results reported here are robust and sound. All data have been properly provided

Annotated reviews are not available for download in order to protect the identity of reviewers who chose to remain anonymous.

---

## Round 0.2 · accepted · Accept

Thank you for addressing the reviewer comments. I believe the manuscript is much improved and does not require further review. Thank you for your important contribution. This manuscript is ready for publication.